# Correlation between Lung Function and Functional Movement in Healthy Adults

**DOI:** 10.3390/healthcare8030276

**Published:** 2020-08-16

**Authors:** Hyun-Seung Kim, Sung-Hyoun Cho

**Affiliations:** 1Department of Medical Sciences, Graduate School of Nambu University, Gwangju 62271, Korea; sang6464@hanmail.net; 2Department of Physical Therapy, Nambu University, Gwangju 62271, Korea

**Keywords:** lung function, functional movement screen, maximum expiratory pressure, maximum inspiratory pressure, inline lunge, trunk stability push-up

## Abstract

It is difficult to determine whether lung function improves by simple abdominal strengthening exercises, and studies on the correlation of lung function and functional movement are insufficient. Therefore; we aimed to identify the correlation between lung function and functional movement. We recruited 204 healthy adults and measured their forced vital capacity; forced expiratory volume in 1 s, maximum voluntary ventilation, maximum expiratory pressure (MEP), and maximum inspiratory pressure (MIP). We also extracted data obtained during functional movements. Differences between lung function and functional movement by gender were determined using independent sample t-tests, while the relationship between lung function and functional movement variables were analyzed using Pearson′s correlation coefficient. Significant gender-based differences between lung function and functional movement, a negative correlation between MIP and inline lunge, and a significantly positive correlation between MIP and trunk stability push-up occurred in males. In females, a positive correlation occurred between MEP and hurdle step, MIP and deep squat, and MIP and hurdle step. Aspects of lung function (MEP and MIP) showed correlations with functional movements. The results showed that lung function and functional movement were correlated, confirming that there is a relationship between lung function and functional movement in healthy adults.

## 1. Introduction

More recently, the amount of exercise and activity performed by people has decreased, while sitting time has increased [1]. As activity decreases, weight rapidly increases, subsequently affecting lung function [2,3]. Although the mechanism for this has not been clarified, several previous studies have reported a correlation between the deterioration of lung function, abdominal obesity, atherosclerosis, and the metabolic syndrome [4,5]. When lung function decreases in people without lung disease, the incidence of vascular disease is reportedly high [6]. Among cardiovascular diseases, coronary artery disease has the highest prevalence and mortality rates [7]. Its most common cause is long-term smoking, which causes fatal lung damage and reduces cardiopulmonary fitness [8]. Coronary artery disease prevalence rates in inactive people are double those in active people [9]. There is a negative correlation between activity and coronary artery disease; cardiorespiratory strength and coronary artery disease are also negatively correlated [9,10].

Cardiopulmonary strength is a physiological indicator of motor performance [11]. It is primarily determined by genetic factors inherited from parents and secondary lifestyles (regular physical activity and exercise) [11,12]. In a preceding study, muscle mass was found to generally improve respiratory performance and have a positive effect on cardiopulmonary function [13]. Conversely, body fat mass is known to have a negative correlation with lung function [14]. Therefore, it is important to provide physical fitness and exercise ability tests to evaluate cardiopulmonary strength.

In addition, flexible movement methods are needed to help improve or maintain the range of motion of each joint in order to maintain functional movement of the body [15]. Flexible exercise offers the advantage of reducing damage to the musculoskeletal system by extending shortened soft tissue for proper functioning of each joint [16]. In order to evaluate the core stability, functional movement evaluation using several tools is used for the functional movement screen (FMS) [17]. The FMS is a functional movement test measurement tool used to evaluate basic motion; it can be used to comprehensively evaluate an individual′s limitations and asymmetries [18,19]. The FMS is based on the motion of intrinsic sensory receptors, mobility, and stability; it is a method for evaluating functional movement limitations and defects, such as left and right imbalances [20].

Developing the core muscles is important for strengthening cardiopulmonary function. The core muscles encourage cooperation between the lateral abdominal muscles and the diaphragm of the thorax, creating a synergistic effect for regulating or responding to abdominal pressure, providing stability to the trunk [21]. The diaphragm and pelvic floor are above and below, respectively [22]. The core muscles (gluteus maximus, femoral quadriceps, oblique muscle, rectus abdominis, and popliteus muscles) play an important role in the stability and postural control of the spine during whole-body exercise with the inner muscles (multifidus, transverse abdominis, pelvic floor muscles, and diaphragm); they also play an important role in lung function [23]. Previous studies have reported that core muscles may affect lung function. In addition, the lower extremity muscles are essential for daily life independence, walking, and lung function [24,25].

The lung function test is used for evaluating indicators that objectively assess the functional aspects of the lung, while the lung capacity test measures the amount of air that can be exhaled after maximum inspiration [26]. Both can help to diagnose various lung diseases, such as chronic obstructive pulmonary disease [27]. Further, breathing is a contributing factor for voice disorders; thus, it is important to also understand breathing as a function [28]. Respiratory diseases may cause voice disorders and vice versa, due to excessive or weak occipital muscle contractions [29]. Therefore, various respiratory diseases can be determined by examining the lung function.

However, it is difficult to determine whether lung function is improved by simple abdominal-strengthening exercises, and studies on the correlation between lung function and functional movement are scarce. Although lung function and functional movement have been studied individually, there have been no integrated studies of both entities. Since FMS findings have shown that lung function is related to the diaphragmatic and abdominal muscles, we aimed to determine the correlation between lung function and functional movement in this study.

## 2. Materials and Methods

### 2.1. Study Design

A cross-sectional study was conducted among healthy adults attending Gwangju Nambu University.

### 2.2. Study Participants

This study was conducted on 204 healthy adults (84 males, 120 females) attending Gwangju Nambu University. The study design was approved by the institutional review board (IRB) of Nambu University, Gwangju City, South Korea (NBU-IRB-1041478-2017-HR-017) and was conducted in accordance with the ethical standards of the Declaration of Helsinki. All participants understood the purpose of the study and signed written informed consent forms. During the course of the study, in line with various health guides’ recommendations, individuals who showed any problems with physical activity were excluded [30]. The participants were notified of the days and times of their evaluation, and were requested to adhere to the following instructions prior to their test: (1) No intense physical activity within 24 h before measurement, (2) no alcohol consumption within 48 h prior to measurement, (3) no consumption of painkillers and anti-inflammatory drugs within 48 h prior to measurement, (4) no consumption of drugs within 12 h prior to measurement, and (5) wearing of appropriate clothing (short pants and T-shirts).

### 2.3. Measurement Methods and Tools

#### 2.3.1. Oxygen Saturation

The pulse oximeter (MD300C26, ChoiceMMed, China), an easy-to-use and simple measurement tool, evaluates oxygen saturation non-invasively. To measure oxygen saturation, the participant was stabilized for 3 min in a sitting position and a pulse oximeter was attached to the middle finger. The average of the values, measured three times in 1 min, was calculated, as per the manufacturer’s instructions [31].

#### 2.3.2. Lung Function

The lung function meter (Pony FX, COSMED, Italy) can be used to measure the amount and speed of air entering and exiting the lungs. In order to accurately measure lung function, a similar measurement method to the American Society of Thoracic and Cardiovascular Surgery (ATS) code of conduct was used [32]. This method followed a procedure similar to that used in a previous study [33]. The process was sufficiently explained and demonstrated to the participants. Each participant’s body was maintained at a 90° angle, with the hip joint in a comfortable sitting position against the back of the chair and feet touching the ground. The measurement was recorded three times to obtain the maximum value. Visual feedback was also used, where measurements were made while viewing the amount of inhalation and exhalation displayed on the graph. The lung function variables assessed included the forced vital capacity (FVC), forced expiratory volume in 1 s (FEV1), FEV1/FVC, maximum voluntary ventilation (MVV), maximum expiratory pressure (MEP), and maximum inspiratory pressure (MIP).

#### 2.3.3. Functional Movement Screen

The FMS is a system used to test functional movement by assessing the limits and asymmetries of an individual’s movements, and for setting values for the joint range of movement, balance, and joint and muscle stability [34,35]. The overall FMS score is between 0 and 21. The functional movement test consists of the deep squat, hurdle step, inline lunge, shoulder mobility, active straight-leg raise, trunk stability push-up, and rotary stability movement. We referred to the method of measuring the FMS as a pre-study, where we photographed the participant’s performance using a video shooting technique to ensure reliability. Using this, the test was conducted on one participant by two raters (physical therapists); the examiner was trained to measure functional movement and check the measurement time [36,37].

A functional movement test was performed according to Cook Gray′s functional movement guidelines, and a total of 3 movements were repeated per test. The tests for evaluating the left and right functions were conducted from the left in accordance with the basic guidelines [20]. The lowest score between the two raters was considered the final score [34,35,38] ( Table 1; Table 2, Figure 1).

### 2.4. Data Analysis

The data from this study were analyzed using IBM SPSS Statistics for Windows, version 25.0 (IBM Corp., Armonk, NY, USA). For the descriptive statistics, the participants’ general characteristics were reported as the mean and standard deviation. Lung function and functional movement variables were examined using Pearson′s correlation coefficient. The statistical significance level (α) was set at 0.05. According to the criteria for interpreting correlation coefficients proposed by Cohen (1988), 0.90 to 1.00, 0.70 to 0.90, 0.40 to 0.70, 0.20 to 0.40, and 0.00 to 0.20 indicated very high, high, definitely, low, and very low correlation, respectively [39].

## 3. Results

### 3.1. Characteristics of the Participants

A total of 204 healthy adults (84 males, 120 females) participated in this study. Their general characteristics are outlined in Table 3. In terms of age, height, body mass index, total body water, soft lean mass, mineral mass, protein mass, weight, skeletal muscle mass, and fat-free mass, male participants had significantly higher values than female participants. Conversely, the body fat mass, body fat percentage, and SpO_2_ of female participants were significantly higher than those of male participants (Table 3).

### 3.2. Analysis of Lung Function by Gender

There were significant gender-related differences in lung function in terms of the FVC, FEV1, MVV, MEP, and MIP (*p* < 0.05). In all lung function variables except for FEV1/FVC, male participants had higher values (Table 4).

### 3.3. Analysis of Functional Movement Screen Scores by Gender

The total FMS score differed significantly at 12.39 ± 2.47 and 11.69 ± 2.46 points for male and female participants, respectively (*p* < 0.05). The deep squat, hurdle step, active straight-leg raises, and trunk stability push-up scores showed significant inter-gender differences (*p* < 0.05) (Table 5).

### 3.4. Correlation between Lung Function Variables and Functional Movement Screen Scores

On evaluating the correlation between lung function and functional motion indicators in both genders, there was a positive correlation between deep squat (r = 0.168 *p* = 0.017) and active straight-leg raise (r = −0.314 *p* = 0.000) in the FVC category; however, trunk stability push-up (r = 0.491 *p* = 0.000) showed a negative correlation (*p* < 0.05). In the FEV1 category, there was a positive correlation with deep squat (r = 0.142 *p* = 0.042), and a negative correlation with active straight-leg raise (r = −0.301 *p* = 0.000). There was a positive correlation between FEV1 and trunk stability push-up (r = 0.469 *p* = 0.000) (*p* < 0.05). In the FEV/FVC category, there was a negative correlation with rotary stability (r = −0.158 r = 0.024) (*p* <0.05). In the MVV category, there was a negative correlation with active straight-leg raise (r = −0.260 p = 0.000) and a positive correlation with trunk stability push-up (r = 0.470 *p* = 0.000) (*p* < 0.05). In the MEP category, there was a positive correlation between hurdle step (r = 0.159 *p* = 0.023) and trunk stability push-up (r = 0.158 *p* = 0.024) (*p* < 0.05). Additionally, in the MIP category, there was a positive correlation with deep squat (r = 0.193 *p* = 0.006) and negative correlation with active straight-leg raise (r = −0.163 *p* = 0.020). In addition, there was a static correlation between the MIP and trunk stability push-up (r = 0.284 *p* = 0.000) (*p* < 0.05) (Table 6).

In male participants, the inline lunge movement (r = −0.237 *p* = 0.030) showed a negative correlation with MIP (*p* < 0.05), while the trunk stability push-up (r = 0.216 *p* = 0.049) showed a positive correlation (*p* < 0.05). In female participants, there was a positive correlation between hurdle steps (r = 0.249 *p* = 0.006) and MEP (*p* < 0.05), and in terms of MIP, a positive correlation was observed with deep squats (r = 0.212 *p* = 0.020) and hurdle steps (r = 0.217 *p* = 0.017) (*p* < 0.05) (Table 7 and Table 8).

## 4. Discussion

The purpose of this study was to identify the association between lung function variables (FVC, FEV1, FEV/FVC, MVV, MEP, and MIP) and functional motion (deep squat, hurdle step, inline lunge, shoulder mobility, active straight-leg raise, trunk stability push-up, and rotary stability).

In terms of the lung function variables in male participants, the MIP and inline lunge showed negative correlations (*p* < 0.05). These results correlated with body stability, showing the relationship between lower-extremity muscles, lung function variables, grip strength, and respiratory muscles. However, in a previous study, a lower correlation coefficient was shown for grip strength because it is influenced by the lower body muscles rather than those of the upper body [40]. Among the lower-extremity muscles, the quadriceps femoris, which plays an important role in daily living ability, is closely related to respiratory function [41].

Weakening of the femoral quadriceps increases movement fatigue. Muscles that are infrequently used cause difficulty in daily life movements and result in the weakness of whole-body muscles [42,43]. Therefore, it is speculated that this negative correlation could indicate excessive use of weakened respiratory muscles during breathing, which in turn affects respiratory function and lung ventilation ability [44]. In our study, there was a positive correlation between trunk stability push-up and MIP (*p* < 0.05). This result is in line with those of previous studies that have shown a positive correlation in functional movement with increased MIP (lung function) [45,46]. The sternocleidomastoid is an upper trunk muscle, which acts as an auxiliary inhalation muscle, lifts the rib cage upward when inhaling, and expands the lung volume. This shows that lung function is increased by movement of the rib cage, which subsequently improves respiratory function [47].

In our study, the MEP showed a positive correlation with hurdle steps (*p* < 0.05) in female participants, while the MIP showed a positive correlation with deep squats and hurdle steps (*p* < 0.05). The stability of the trunk is achieved through proper activation and coordination of the abdominal muscles and the spinal vertebral muscles, because they are used together with the diaphragm, intercostal muscles, and quadriceps, which act as the main muscles during breathing [48]. An increase in interbody muscle strength showed a positive correlation with the increase in MEP and MIP at expiration and inspiration, respectively [49].

When the lower extremity moves, the deep muscles in the lumbar region proactively adjust for anticipatory postural control, and the activity of the abdominal muscles vary depending on the direction of the changed movement towards the lower extremity [50]. Previous studies have shown that changes in the thickness of the abdominal muscles during the adductor movement of the lower extremities occurred significantly when the abdominal muscles contracted. The mechanism involved is as follows: During internal rotation of the hip joint, the vertebral and pelvic muscles contract simultaneously, the transverse abdominis and multifidus muscles contract beforehand, while the transversus abdominis stops at the iliac spine and appears to pull the long bone in a transverse direction towards the spine. Therefore, by correctly aligning the spine to complement the structural instability of the spine [51], the body is stabilized by overall muscle contraction of the abdominal muscles, which is related to active breathing [52,53].

Previous studies have shown that balance is increased by increasing exercise efficiency, increasing physical activity, improving respiratory muscle and lung function [54], controlling the lumbar region [55], and improving the abdominal muscle strength [56,57]. Therefore, the lower abdomen and hip flexion occur organically, and the hip muscles are related to the balance of the body [58]. In our study, we found that muscle strength in a healthy adult is related to respiratory function, and the lower-extremity muscle strength used for functional movement is a basic factor for independent daily living, due to the correlation between MEP and MIP. In addition, the breathing pattern of healthy adults is related to improved muscle strength and coordination of the breathing muscles in terms of functional exercise, and to the stability and breathing ability of the trunk. Therefore, it may be concluded that the muscle strength of the limbs is related to the muscle strength of the trunk and to lung function and respiratory muscle movement.

This study had some limitations. First, electromyography (EMG) assessment was not performed in the healthy adults; hence, the activation of muscles could not be verified. In addition, we could not identify any causal relationships because we only investigated the correlation between lung function and functional movement. Based on previous studies and hypotheses, we believe that it is necessary to verify this mechanism through EMG. Additionally, when we attempted to compare FVC, FEV1, FEV/FVC, MVV, MEP, and MIP values with those of previous studies, we found a lack of references regarding the link between lung function and functional movement. Further research is needed to quantitatively evaluate whether lung function or the respiratory muscles increase with increasing functional movement.

## 5. Conclusions

The purpose of this study was to investigate whether there is a correlation between lung function and functional movement in 204 healthy adults. The results showed that lung functional parameters (FVC, FEV1, FEV/FVC, MVV, MEP, and MIP) were correlated with functional movement (the deep squat, hurdle step, inline lunge, shoulder mobility, active straight-leg raise, trunk stability push-up, and rotary stability). These findings confirm that there is a relationship between lung function and functional movement in healthy adults. By developing effective muscle-strengthening programs for the pulmonary function necessary for activities of daily living, we believe that future studies related to pulmonary function would need to be approached in various ways. Our results are expected to be useful as reference values when measuring lung function in healthy adults.

## Figures and Tables

**Figure 1 healthcare-08-00276-f001:**
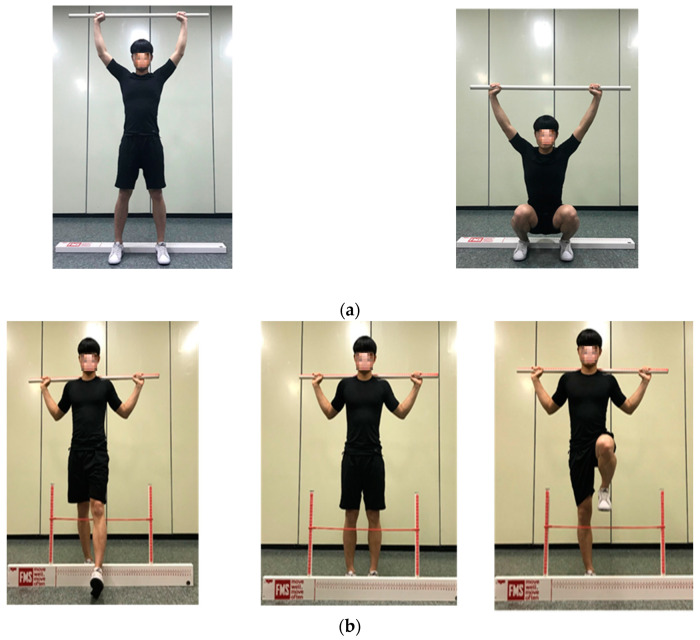
Functional movement screen test: (**a**) deep squat; (**b**) hurdle step; (**c**) inline lunge; (**d**) shoulder mobility; (**e**) active straight-leg raise; (**f**) trunk stability push-up; (**g**) rotary stability.

**Table 1 healthcare-08-00276-t001:** Functional movement screen measurement variables and evaluated content.

Measuring Variable	Assessment
Deep Squat	Assessed the bilateral, symmetrical, and functional mobility of the hips, knees, and ankles. Overhead, it assessed the bilateral and symmetrical mobility of the shoulders and thoracic spine.
Hurdle Step	Assessed the bilateral functional mobility and stability of the hips, knees, and ankles.
Inline Lunge	Assessed the hip and ankle mobility and stability, quadriceps flexibility, and knee stability.
Shoulder Mobility	Assessed the bilateral range of motion of the shoulder by combining internal rotation with adduction and external rotation with abduction.
Active Straight-Leg Raise	Assessed active hamstring and gastric-soleus flexibility while maintaining a stable pelvis and active extension of the opposite leg.
Trunk Stability Push-up	Assessed trunk stability in the sagittal plane using a symmetrical upper-extremity motion.
Rotary Stability	Assessed multi-plane trunk stability using a combined upper- and lower-extremity motion.

**Table 2 healthcare-08-00276-t002:** Scores of the contents in the Functional movement screen evaluation.

Score	Contents
3	Complete motion pattern matching the action
2	Matching but incorrect form of motion pattern due to compensatory action
1	Inconsistent and incomplete motion pattern
0	Occurrence of pain

**Table 3 healthcare-08-00276-t003:** General characteristics of the participants.

Variables	Participants (*n* = 204)	Male (*n* = 84)	Female (*n* = 120)	*t*	*p*
Age (year)	27.0 ± 23.06	23.86 ± 1.89	22.51 ± 1.23	5.746	<0.001
Height (cm)	187.0 ± 166.07	173.87 ± 5.17	160.62 ± 5.08	18.193	<0.001
Body mass index (kg/m^2^)	41.40 ± 23.61	24.21 ± 3.90	23.19 ± 4.41	1.715	0.088
Total body water (L)	51.30 ± 34.37	41.99 ± 4.95	29.03 ± 3.73	20.286	<0.001
Soft lean mass (kg)	66.0 ± 44.22	54.09 ± 6.39	37.30 ± 4.81	20.378	<0.001
Mineral mass (kg)	5.15 ± 3.32	3.97 ± 0.54	2.87 ± 0.36	16.242	<0.001
Protein mass (kg)	14.20 ± 9.29	11.43 ± 1.37	7.79 ± 1.01	20.607	<0.001
Body fat (kg)	50.80 ± 18.41	15.91 ± 7.61	20.16 ± 8.15	−3.768	<0.001
Weight (kg)	108.10 ± 65.39	73.29 ± 12.59	59.86 ± 12.05	7.687	<0.001
Skeletal muscle mass (kg)	40.50 ± 26.03	32.49 ± 4.14	21.51 ± 3.07	20.670	<0.001
Body fat mass (kg)	50.80 ± 18.41	15.91 ± 7.61	20.16 ± 8.15	−3.768	<0.001
Fat-free mass (kg)	70.40 ± 46.98	57.38 ± 6.84	39.70 ± 5.09	20.111	<0.001
Body fat percentage (%)	51.70 ± 27.81	20.89 ± 6.68	32.65 ± 6.74	−12.304	<0.001
SpO_2_ (%)	100.00 ± 98.23	97.80 ± 0.93	98.53 ± 1.17	−5.006	<0.001

SpO_2_, Saturation of percutaneous oxygen.

**Table 4 healthcare-08-00276-t004:** Analysis of lung function variables by gender.

	Males	Females	*t*	*p*
FVC	4.64 ± 0.59	3.27 ± 0.44	18.193	<0.001 *
FEV1	3.95 ± 0.48	2.82 ± 0.37	17.849	<0.001 *
FEV1/FVC	85.03 ± 6.02	86.70 ± 6.56	−1.848	0.066
MVV	155.18 ± 31.43	99.96 ± 20.77	14.093	<0.001 *
MEP	38.80 ± 12.99	34.63 ± 10.90	2.480	0.014 *
MIP	45.26 ± 17.08	37.23 ± 11.08	3.787	<0.001 *

* *p* < 0.05 FVC, forced vital capacity; FEV1, forced expiratory volume in 1 s; FEV1/FVC, forced expiratory volume in 1 s/ forced vital capacity; MVV, maximum voluntary ventilation; MEP, maximum expiratory pressure; MIP, maximum inspiratory pressure.

**Table 5 healthcare-08-00276-t005:** Analysis of functional movement screen scores by gender.

	Males	Females	*t*	*p*
Deep squat	1.89 ± 0.58	1.69 ± 0.62	2.369	0.019 *
Hurdle step	1.65 ± 0.57	1.47 ± 0.55	2.370	0.019 *
Inline lunge	1.71 ± 0.59	1.72 ± 0.54	−0.030	0.976
Shoulder mobility	2.29 ± 0.82	2.45 ± 0.75	−1.481	0.140
Active straight Leg raise	1.29 ± 0.55	34.92 ± 11.19	−6.219	<0.001 *
Trunk stability push up	2.00 ± 0.98	1.00 ± 0.53	8.496	<0.001 *
Rotary stability	1.61 ± 0.54	1.51 ± 0.53	1.296	0.196
FMS total score	12.39 ± 2.47	11.69 ± 2.46	1.999	0.047 *

* *p* < 0.05 FMS, functional movement screen.

**Table 6 healthcare-08-00276-t006:** Correlation between lung function and functional movement screen scores in the entire cohort.

		Deep Squat	Hurdle Step	Inline Lunge	Shoulder Mobility	Active Straight-Leg Raise	Trunk Stability Push-Up	Rotary Stability
FVC	r	0.168	0.126	0.009	−0.123	−0.314	−0.491	0.089
	(*p*)	(0.017 *)	(0.073)	(0.897)	(0.079)	(0.000 *)	(0.000 *)	(0.205)
FEV1	r	0.142	0.119	−0.020	−0.107	−0.301	0.469	0.037
	(*p*)	(0.042 *)	(0.090)	(0.779)	(0.126)	(0.000 *)	(0.000 *)	(0.596)
FEV/FVC	r	−0.071	−0.023	−0.048	0.044	0.085	−0.094	−0.158
	(*p*)	(0.316)	(0.743)	(0.500)	(0.531)	0(.226)	(0.181)	(0.024 *)
MVV	r	0.109	0.125	0.004	−0.046	−0.260	0.470	0.037
	(*p*)	(0.119)	(0.075)	(0.957)	(0.515)	(0.000 *)	(0.000 *)	(0.596)
MEP	r	0.127	0.159	−0.001	−0.034	−0.096	0.158	−0.025
	(*p*)	(0.070)	(0.023*)	(0.985)	(0.626)	(0.172)	(0.024 *)	(0.719)
MIP	r	0.193	0.100	−0.066	−0.035	−0.163	0.284	0.009
	(*p*)	(0.006 *)	(0.154)	(0.349)	(0.614)	(0.020 *)	(0.000 *)	(0.902)

* *p* < 0.05, r = Pearson correlation coefficient FVC, forced vital capacity; FEV1, forced expiratory volume in 1 s; FEV1/FVC, forced expiratory volume in 1 s/ forced vital capacity; MVV, maximum voluntary ventilation; MEP, maximum expiratory pressure; MIP, maximum inspiratory pressure.

**Table 7 healthcare-08-00276-t007:** Correlation between lung function variables and functional movement screen scores in male participants.

		Deep Squat	Hurdle Step	Inline Lunge	Shoulder Mobility	Active Straight-Leg Raise	Trunk Stability Push-Up	Rotary Stability
FVC	r	0.128	−0.033	0.016	−0.121	0.046	0.203	0.094
	(*p*)	(0.245)	(0.767)	(0.888)	(0.274)	(0.681)	(0.064)	(0.394)
FEV1	r	0.082	−0.056	−0.075	−0.095	0.102	0.158	−0.001
	(*p*)	(0.456)	(0.615)	(0.497)	(0.388)	(0.354)	(0.152)	(0.995)
FEV/FVC	r	−0.033	0.005	−0.096	0.032	0.122	−0.027	−0.139
	(*p*)	(0.764)	(0.963)	(0.387)	(0.773)	(0.270)	(0.808)	(0.208)
MVV	r	−0.006	−0.006	−0.124	−0.020	0.016	0.150	−0.124
	(*p*)	(0.954)	(0.957)	(0.262)	(0.859)	(0.883)	(0.174)	(0.262)
MEP	r	0.055	0.002	−0.153	−0.150	−0.098	0.118	−0.115
	(*p*)	(0.622)	(0.987)	(0.164)	(0.172)	(0.375)	(0.285)	(0.298)
MIP	r	0.112	−0.082	−0.237	−0.083	−0.155	0.216	−0.086
	(*p*)	(0.310)	(0.457)	(0.030 *)	(0.451)	(0.158)	(0.049 *)	(0.438)

* *p* < 0.05, r = Pearson correlation coefficient. FVC, forced vital capacity; FEV1, forced expiratory volume in 1 s; FEV1/FVC, forced expiratory volume in 1 s / forced vital capacity; MVV, maximum voluntary ventilation; MEP, maximum expiratory pressure; MIP, maximum inspiratory pressure.

**Table 8 healthcare-08-00276-t008:** Correlation between lung function variables and functional movement screen scores in female participants.

		Deep Squat	Hurdle Step	Inline Lunge	Shoulder Mobility	Active Straight-Leg Raise	Trunk Stability Push-Up	Rotary Stability
FVC	r	0.008	0.010	0.021	−0.015	−0.061	−0.072	−0.034
	(*p*)	(0.927)	(0.913)	(0.823)	(0.867)	(0.507)	(0.437)	(0.708)
FEV1	r	−0.030	0.014	0.015	0.011	−0.060	−0.093	−0.111
	(*p*)	(0.749)	(0.879)	(0.872)	(0.905)	(0.516)	(0.311)	(0.229)
FEV/FVC	r	−0.061	−0.007	−0.015	0.031	0.005	−0.033	−0.155
	(*p*)	(0.507)	(0.943)	(0.870)	(0.738)	(0.959)	(0.724)	(0.091)
MVV	r	−0.021	0.022	0.161	0.117	0.033	0.066	0.043
	(*p*)	(0.820)	(0.810)	(0.079)	(0.204)	(0.718)	(0.471)	(0.645)
MEP	r	0.140	0.249	0.139	0.103	0.000	0.020	0.019
	(*p*)	(0.127)	(0.006 *)	(0.131)	(0.263)	(0.999)	(0.827)	(0.834)
MIP	r	0.212	0.217	0.130	0.081	−0.011	0.067	0.056
	(*p*)	(0.020 *)	(0.017 *)	(0.158)	(0.380)	(0.907)	(0.469)	(0.540)

* *p* < 0.05, r = Pearson correlation coefficient FVC, forced vital capacity; FEV1, forced expiratory volume in 1 s; FEV1/FVC, forced expiratory volume in 1 s/ forced vital capacity; MVV, maximum voluntary ventilation; MEP, maximum expiratory pressure; MIP, maximum inspiratory pressure.

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
