# Peer review of "Correlation between Lung Function and Functional Movement in Healthy Adults"

_healthcare, 2020, doi:10.3390/healthcare8030276_

Round 1

Reviewer 1 Report

This was a fairly simple study correlating lung function to functional movement. The main problem is that a correlation cannot infer cause. The authors do however infer cause (see line 237-239).

Author Response

We attached the response to reviewer file.

Reviewer 2 Report

The following article is very interesting, considering the current situation worldwide. It is demonstrated that some of the major sequels of COVID-19 are respiratory. So knowing the correlation between lung function and functional movement is important.

However, some comments are made below in relation to the article.

Maybe it is not necessary the Figure 1, 2 and 3. They do not provide additional information.

It is the same for the figure 4, it does not provide additional information.

In relation to the FMS test, it is recommendable more information about the experience of the evaluator/s measuring the test and how they do. The creators of the FMS explain that an evaluator needs more than 100 measurements to be well the measurements. It is one of the biggest problems with these tests.

Table 1 and 2 is it identical from the authors of the FMS or other references? In this case, should be indicated the reference.

In the point 2, Materials and Methods I cannot see the anthropometry measurements, should be information about them.

It is sure, that there are differences between genders. Why all the measurements have done by gender? In my opinion should be a column with data from the whole group.

Author Response

(The authors gave the same response as above.)

Round 2

Reviewer 1 Report

My main concern previously was that this was simply a correlational study and the authors appeared anxious to attribute causality. This remains my concern. While the authors revised one paragraph, other sentences remain that imply causation. For example, L 25-27 and L 263-265. The authors would need to correct this and also make it clear that the study design was a limitation. 

Author Response

We attached the file.

Reviewer 2 Report

Thank you very much for taking into account the recommendations to improve the document. As I said last time, the following article is very interesting, considering the current situation worldwide. It is demonstrated that some of the major sequels of COVID-19 are respiratory. So knowing the correlation between lung function and functional movement is important. After reviewing the manuscript, just a small error in the “Body Mass Index (“ in Table 3.

Author Response

We attached the file.
